# On Correlation Functions as Higher-Spin Invariants

**Adrien Scalea**

Physique de l'Univers, Champs et Gravitation, Université de Mons—UMONS, 20 Place du Parc,
B-7000 Mons, Belgium; adrien.scalea@student.umons.ac.be

**Abstract:** (Chern–Simons) vector models exhibit an infinite-dimensional symmetry, the slightly-broken higher-spin symmetry with the unbroken higher-spin symmetry being the first approximation. In this note, we compute the $n$-point correlation functions of the higher-spin currents as higher-spin invariants directly on the CFT side, which complements earlier results that have a holographic perspective.

**Keywords:** higher spin; correlation function; vector model

## 1. Introduction

(Chern–Simons) vector models are a rich class of three-dimensional conformal field theories that can be of interest for a number of reasons. Firstly, these CFTs describe critical behaviour of many physical systems, e.g., the famous Ising model, which can be realised as the $O(N)$-vector model, describes the critical behaviour of the $O(N)$-magnetic at the Curie point. Secondly, a hypothetical bulk AdS/CFT dual of these CFTs [1–4] gives a class of higher-spin gravities, the latter can, at least formally, be defined by inverting the correlation functions [5–8]. Thirdly, Chern–Simons vector models were conjectured to exhibit a number of remarkable dualities [9–14], of which, in this work, we concentrate on the three-dimensional bosonisation duality.

The simplest gauge-invariant operators in Chern–Simons vector models are higher-spin currents, that are operators of type $J_s = \bar{\phi} D \ldots D \phi + \ldots$ or $J_s = \bar{\psi} \gamma D \ldots D \psi + \ldots$, depending on whether the model's matter is bosons $\phi$ or fermions $\psi$. The term 'higher-spin current' is jargon. The higher-spin currents are not conserved unless we deal with a free or very large-$N$ vector model and even in this case they are conserved tensors for $s > 1$ rather than currents (to obtain a current, one needs to contract it with a conformal Killing tensor). In addition, $s = 0$ and $s = 1$, whenever present, are included into the multiplet of higher-spin currents. We will, however, stick with this unfortunate terminology. Higher-spin currents also turn out to be single-trace operators from the holographic perspective, and are dual to massless higher-spin fields in $AdS_4$. To prove the $3d$ bosonisation duality, it is sufficient to show that all $n$-point correlation functions of the dual theories are the same, provided we relate the free parameters appropriately. Therefore, we can concentrate on the higher-spin currents and ignore all other local gauge-invariant operators.

In the very large-$N$ limit, the higher-spin currents are conserved. Via the Noether theorem, they lead to an infinite-dimensional extension of the conformal symmetry $so(3,2)$, with the spin-two current, the stress-tensor, manifesting $so(3,2)$. The resulting algebra is usually called the higher-spin algebra and it is also the symmetry algebra of the free boson's and free fermion's equations of motion, e.g., refs. [15–20]. Historically, it was identified [15] as the even subalgebra of the Weyl algebra $A_2$, which is the algebra of observables of the $2d$ harmonic oscillator.

The (unbroken) higher-spin symmetry is the usual symmetry, i.e., there is a Lie algebra acting on the physical spectrum of operators. Interestingly, the free matter fields and the multiplet of the higher-spin currents are the simplest representation of the higher-spin algebra [15,21,22]. The algebra admits an invariant trace $\mathrm{tr}[a \star b - b \star a] = 0$, and the series of invariants

$$\langle J \ldots J \rangle = \text{tr}[\Psi \star \ldots \star \Psi]$$

computes the correlation functions, provided the wave-functions $\Psi$ are chosen wisely. Calculations of this kind were performed in [23–26], where $\Psi$ was taken to represent a multiplet of massless higher-spin fields in $AdS_4$. It is important that the unbroken higher-spin symmetry is a signature of the free CFT's behaviour in $d \geq 3$ [27–30]. The results of [27] require finite $N$, but the higher-spin currents are conserved at $N = \infty$ as well. Importantly, they are also conserved at $N = \infty$ in the interacting vector models. Uniqueness of this type of invariants can also be shown [31].

When interactions are turned on, either directly or by departing from the very large-$N$ limit, the higher-spin currents cease to be conserved, except for the stress-tensor, $s = 2$, and for the global symmetry current, $s = 1$. As a simple consequence of the smallness of operators' spectrum of vector models, the non-conservation operator has a very restricted form of a composite operator, built of $J$s themselves [9,10]. As a result, the non-conservation is still very useful to impose on correlation functions [10] and this type of symmetry breaking was dubbed slightly-broken higher-spin symmetry in [10]. Mathematically, the slightly-broken higher-spin symmetry is not a symmetry. It is not realised as an action of some Lie algebra on a multiplet of operators. However, it can be understood as a strong homotopy algebra, that deforms the action of the higher-spin algebra [32–34].

While Chern–Simons vector models have simple actions, the (slightly-broken) higher-spin symmetry is by far the most efficient way to compute correlation functions of higher-spin currents in vector models, e.g., refs. [35–42]. For free or very large-$N$ vector models, this calculation was performed in [23–26,43] (note that the calculation of [43] applied a regularisation that effectively replaced the ill-defined vertices [44,45] with the higher-spin invariant) with an additional proviso of identifying the correlation functions with the invariants of the higher-spin algebra. The correlators can also be computed via the textbook Wick contractions [46] of free fields.

The main point of the present note is to exclude the holographic aspect present in [23–26,43]. In other words, in this note we adopt a purely CFT view on the higher-spin symmetry. The wave-functions of [23–26,43] represent a multiplet of higher-spin fields that are duals of higher-spin currents. Due to the fact that the bulk dual of Chern–Simons vector models is not known, and is unlikely to exist as a reasonably local field theory, one cannot just extract the Chern–Simons vector models' correlators from holography. Fortunately, the key features of the higher-spin symmetry (such as mixing spins and derivatives) that generically invalidate the field theory approach are harmless on the CFT side. It should be mentioned that the dual theory has a closed local subsector [47–51], which is an $AdS_4$-deformation of the chiral higher-spin gravity in flat space [52–59], see, e.g., ref. [60] for more on higher-spin gravities. It would be interesting to compute holographic correlation functions in this model, but we believe it can be achieved more efficiently on the CFT side. We hope that this is a useful first step in the programme of computing correlation functions of higher-spin currents in Chern–Simons vector models, as invariants of the slightly-broken higher-spin symmetry.

The note is organised as follows. In Section 2 we recall the results of [61] on the general structure of 3$d$ conformal correlators. In Section 3 we define the wave-functions that represent a generating function of higher-spin currents and compute the higher-spin invariants. Two appendices collect some useful identities and definitions.

## 2. Structure of Correlation Functions in Three Dimensions

In three dimensions, one has the isomorphism $so(2,1) \simeq sl(2,\mathbb{R})$, implying that a traceless rank-$s$ Lorentz tensor can be represented by a rank-2$s$ spin-tensor. In Appendix A, we detail notations and conventions, but in brief, we note that a 3-vector $\mathbf{x}^m$ ($m = 0,1,2$) can be mapped into a symmetric bi-spinor $X^{\alpha\beta}$ ($\alpha, \beta = 1,2$).

Higher-spin currents are symmetric and traceless tensors $J_{a_1...a_s}(\mathbf{x})$. In addition, they are conserved $\partial^b J_{ba_2...a_s} = 0$, so that they are primary fields. Thanks to the isomorphism, they are mapped to $J^{\alpha_1...\alpha_{2s}}(X)$ and one can pack them into a generating function $j(X, \eta) = J^{\alpha_1...\alpha_{2s}}(X) \eta_{\alpha_1} ... \eta_{\alpha_{2s}}$, where $\eta_\alpha$ is an auxiliary polarisation spinor. The conservation implies

$$\frac{\partial}{\partial X^{\alpha\beta}} \frac{\partial^2}{\partial \eta_\alpha \partial \eta_\beta} j(X, \eta) = 0 \,. \tag{1}$$

It turns out, that conformally invariant correlation functions of tensor operators can depend on very few atomic conformally invariant structures [61]. There are two parity-even atomic structures

$$P_{ij} = \eta_i^\alpha \eta_j^\beta \left(X_{ij}^{-1}\right)_{\alpha\beta}, \qquad Q_{j,k}^i = \eta_i^\alpha \eta_i^\beta \left(X_{ji}^{-1} - X_{ki}^{-1}\right)_{\alpha\beta}, \tag{2}$$

where $X_{ij}^{-1} \equiv \left(X_{ij}\right)^{-1} \equiv \left(X_i - X_j\right)^{-1}$. We have $P_{ij} = -P_{ji}$ and $Q_{j,k}^i = -Q_{k,j}^i$. Defining the inversion map as

$$R\eta_\alpha^i = -\frac{X_{i\,\alpha}{}^\beta \eta_\beta^i}{|X_i|} = (X_i^{-1})_\alpha{}^\beta \eta_\beta^i, \tag{3}$$

we observe $RP_{ij} = P_{ij}$ and $RQ_{j,k}^i = Q_{j,k}^i$, which proves the structures to be parity-even. There is also one parity-odd invariant structure

$$S_{jk}^i = \frac{\eta_k^\alpha \left(X_{ik} X_{ij}\right)_{\alpha\beta} \eta_j^\beta}{x_{ij} x_{ik} x_{jk}} \,, \tag{4}$$

where $x_{ij} \equiv |\mathbf{x}_{ij}| = \sqrt{-|X_{ij}|}$. Conformally invariant correlation functions depend on the cross-ratios, and are polynomials in $P$s, $Q$s, and $S$s. The exponents of $P$s, $Q$s, and $S$s are constrained by the spin of the operators in an obvious way. For example, the simplest two- and three-point correlators

$$\langle j_s(X_1, \eta_1) j_s(X_2, \eta_2) \rangle \sim \frac{1}{x_{12}^2} (P_{12})^{2s} \,, \tag{5}$$

$$\langle j_{s_1}(X_1, \eta_1) j_0(X_2) j_0(X_3) \rangle \sim \frac{1}{x_{12} x_{23} x_{31}} (Q_1)^{s_1} \,, \tag{6}$$

where $j_s$ is a conserved higher-spin current and $j_0$ is a scalar operator of dimension 1.

## 3. Correlation Functions as Higher-Spin Invariants

In this section, we first recall the definition of the relevant higher-spin algebra, together with the star-product, as a convenient tool to work with it. We also introduce a conformally friendly basis for the generators. Next, we fix the form of the wave-functions $\Psi$ and compute the correlation functions.

### 3.1. Higher-Spin Algebra

The isomorphism $so(3, 2) \simeq sp(4, \mathbb{R})$, allows us to use the $sp(4)$ generators $T_{AB} = T_{BA}$, $A, B = 1, ..., 4$, such that

$$[T_{AB}, T_{CD}] = T_{AC} \epsilon_{BD} + 3 \text{ terms}, \tag{7}$$

where $\epsilon_{AB} = -\epsilon_{BA}$ and $\epsilon_{AB} \epsilon^{AC} = \delta_B{}^C$. With the four operators $Y_A$ satisfying the canonical commutation relations $[Y_A, Y_B] = 2i\epsilon_{AB}$, one can realise the above commutation relations as $T_{AB} = \frac{-i}{4} \{Y_A, Y_B\}$, which is the standard oscillator realisation. The associative algebra of functions $f(Y)$ in $Y^A$ is the Weyl algebra $A_2$ (the subscript 2 is the number of canonical pairs). Its even subalgebra $A_2^e$ of functions $f(Y) = f(-Y)$ is the higher-spin algebra we need. We can also split $Y_A = (y_\alpha, \bar{y}_\alpha)$ and $\epsilon_{AB} = \text{diag}(\epsilon_{\alpha\beta}, \epsilon_{\alpha\beta})$, with $\epsilon_{12} = 1$.

More abstractly, a higher-spin algebra can be defined, for any irreducible representation of the conformal algebra, as the quotient of the universal enveloping algebra by a two-sided ideal that is the annihilator of this representation or, in the field theory language, as the symmetry algebra of the corresponding conformally invariant field equation [19]. An important fact, is that for the free fermion and free boson representations, this ideal gets resolved by the oscillator realisation. Another important fact for the $3d$ bosonisation duality to take place, is that the higher-spin algebras of the free fermion and the free boson are isomorphic to the same $A_2^e$, which is explicit already in [15].

Higher-spin algebras turn out to be infinite-dimensional associative algebras that contain the conformal algebra as a Lie subalgebra (any associative algebra leads to a Lie algebra where the Lie bracket is defined as the commutator). Therefore, any higher-spin algebra can be viewed as an infinite-dimensional extension of the conformal symmetry.

### 3.1.1. Star-Product

Instead of working with an algebra of operators, it is convenient to use the algebra of functions in commuting variables $Y_A$ (symbols) with the (associative) Moyal–Weyl star-product. It admits an integral form and a more standard differential form (simple $(2\pi)^{-4}$ prefactor is omitted or included into the definition of $\int$ below)

$$f(Y) \star g(Y) = \int \mathrm{d}^4 U \, \mathrm{d}^4 V \, f(Y+U) g(Y+V) \, e^{iV^A U_A} = f(Y) \exp\left[ i \overleftarrow{\partial}_A \epsilon^{AB} \overrightarrow{\partial}_B \right] g(Y) \, . \quad (8)$$

We will also have to go outside of the space of polynomial functions, e.g., admitting delta function $\delta^2(y) = \int \mathrm{d}^2 s \, e^{i s^\alpha y_\alpha}$. With the star-product, we have $[Y_A, Y_B]_\star := Y_A \star Y_B - Y_B \star Y_A = 2i\epsilon_{AB}$ and the unit element is 1, i.e., $f \star 1 = 1 \star f = f$. We also find useful relations

$$Y_A \star f(Y) = Y_A f + i \frac{\partial f}{\partial Y^A} \, , \qquad f(Y) \star Y_A = Y_A f - i \frac{\partial f}{\partial Y^A} \, . \quad (9)$$

The even subalgebra of the Weyl algebra admits an invariant trace operation, which in terms of symbols $f(Y)$ reads

$$\mathrm{tr}\big(f(Y)\big) = f(0) \quad (10)$$

such that $\mathrm{tr}(f \star g) = \mathrm{tr}(g \star f)$.

### 3.1.2. Conformally Adapted Basis

In view of the CFT nature of the problem, it is convenient to split $T_{AB}$ in such a way as to make the standard basis of conformal generators explicit, e.g., refs. [16,62]. We define $y_\alpha^- = \frac{1}{2}(\bar{y}_\alpha - i y_\alpha)$ and $y^{+\alpha} = \frac{1}{2}(y^\alpha - i \bar{y}^\alpha)$ that obey $[y_\alpha^-, y^{+\beta}]_\star = \delta_\alpha{}^\beta$, which implies that $y_\alpha^\pm$ are the standard creation/annihilation operators. Indeed, the conformal generators read [62]

$$\begin{aligned} P_{\alpha\beta} &= i y_\alpha^- y_\beta^- \, , & K^{\alpha\beta} &= -i y^{+\alpha} y^{+\beta} \, , \\ D &= \tfrac{1}{2} y^{+\alpha} y_\alpha^- \, , & L^\alpha{}_\beta &= y^{+\alpha} y_\beta^- - \tfrac{1}{2} \delta_\beta^\alpha y^{+\gamma} y_\gamma^- \, . \end{aligned} \quad (11)$$

With the reality conditions $(y_\alpha^-)^\dagger = y^{+\alpha}$, one has $D^\dagger = D$, $(L^\alpha{}_\beta)^\dagger = L_\beta{}^\alpha$ and $P_{\alpha\beta}^\dagger = K^{\alpha\beta}$. The mass-shell condition is manifest since $P^2 = 0$. The basic star-product relations (9) in terms of $y^\pm$ read

$$y_\alpha^\pm \star f(y^+, y^-) = y_\alpha^\pm + \tfrac{1}{2} \partial_\alpha^\mp f \, , \qquad f(y^+, y^-) \star y_\alpha^\pm = y_\alpha^\pm - \tfrac{1}{2} \partial_\alpha^\mp f \, . \quad (12)$$

As a result, $[y_\alpha^\pm, f]_\star = \partial_\alpha^\mp f$ and

$$[y_\alpha^a y_\beta^b, f]_\star = y_\alpha^a \partial_\beta^{\bar{b}} f + y_\beta^b \partial_\alpha^{\bar{a}} f \, , \qquad a, b \in \{+, -\} \, , \quad \bar{a} \equiv -a \, . \quad (13)$$

The action of the conformal generators (11) reads

$$
\begin{aligned}
[P_{\alpha\beta}, f]_\star &= +i\big(y_\alpha^- \partial_\beta^+ + y_\beta^- \partial_\alpha^+\big)f\,, \\
[K_{\alpha\beta}, f]_\star &= -i\big(y_\alpha^+ \partial_\beta^- + y_\beta^+ \partial_\alpha^-\big)f\,, \\
[D, f]_\star &= \tfrac{1}{2}\big(y^{+\alpha}\partial_\alpha^+ - y^{-\alpha}\partial_\alpha^-\big)f\,, \\
[L_{\alpha\beta}, f]_\star &= \Big(y_\alpha^+ \partial_\beta^+ + y_\beta^- \partial_\alpha^- + y_\beta^+ \partial_\alpha^+ + y_\alpha^- \partial_\beta^-\Big)f\,.
\end{aligned}
\tag{14}
$$

We observe that $D$ counts the number of $y^+$ minus the number of $y^-$.

*3.2. Wave-Functions*

Given that the higher-spin algebra is an infinite-dimensional extension of the conformal algebra, it should not be surprising that correlation functions of the higher-spin currents can be computed as simple invariants of this algebra [23–26]. The invariants know nothing about correlation functions per se and must be fed with appropriate wave-functions $\Psi$, that contain the information about the operators' positions and spins. In [23–26], $\Psi$ was defined to reside in $AdS_4$ and it represents a collection of massless fields in $AdS_4$. In addition, $\Psi$ of [23–26] does not transform in the adjoint representation. Below, we fix the form of $\Psi$ on the CFT side, which is the main difference compared to [23–26]. We will find that the wave-function $\Psi$ is simpler than its $AdS_4$ cousin.

Wave-Functions' Properties

The main building block of correlation functions of higher-spin currents is

$$
O_n = \mathrm{tr}\big(\Psi_1 \star \cdots \star \Psi_n\big)\,.
\tag{15}
$$

It is invariant under higher-spin transformations (hence, conformally invariant as well) provided that $\Psi_i \equiv \Psi(X_i, \eta_i|Y)$ transforms in the adjoint representation of the higher-spin algebra, i.e., $\delta_\xi \Psi = [\Psi, \xi]_\star$. To relate this observable to higher-spin currents, we also need to make sure that $\Psi$ obeys the conservation condition (1).

Concerning (15), it is worth noting that it has only cyclic symmetry, which is exactly the symmetry of the correlation functions in vector models with (leftover) global symmetries that are not gauged via the Chern–Simons term. Indeed, if there is a global symmetry, say $U(M)$, the higher-spin currents have a pair of indices $J_i{}^k$. Correlation functions of such currents have only cyclic symmetry. In some sense, (15) is the master higher-spin invariant and all the others can be obtained by projecting it (say on the bosonic currents, since by default it contains the super-currents as well) and symmetrising over the external legs.

In [23–26], the authors used the AdS/CFT formalism, where all physical information is encoded in the master field $B(X, z, \eta|Y)$ ($z$ is the radial coordinate on $AdS_4$). This field transforms in a twisted-adjoint representation. However, one can build $\Psi = B \star \delta^2(y)$ that transforms in the adjoint one, which still resides in the bulk. To give an idea of the functional class used for the holographic calculations in [23–26], the main building block of $B$ was found to be

$$
\Phi(F, \xi, \theta) = K \exp i(-yF\bar{y} + \xi y)\,,
\tag{16}
$$

where $K$ is the scalar boundary-to-bulk propagator. Matrix $F$, spinor $\xi$, and $K$ depend on the bulk and boundary coordinates. Explicit formulas can be found in [23–26]. Let us note that $B$ does not have any obvious boundary limit.

In order to find the wave-function $\Psi(X, \eta|y^\pm)$ directly on the CFT side, we will use the following physical conditions. (1) $\Psi$ must be a generating function of quasi-primary operators at $X = 0$; (2) $\Psi$ must satisfy the equation of motion—covariant constancy condition, which reconstructs $X$-dependence; (3) it has to be a generating function of conserved higher-spin currents. We should also take into account that there is no unique solution $\Psi$ that satisfies (1–3), since we can always rescale any spin-$s$ component by some numerical factor.

The first condition can be translated into a simple equation

$$[K_{\alpha\beta}, \Psi(0, \eta | y^{\pm})]_{\star} = (y_{\alpha}^{+} \partial_{\beta}^{-} + y_{\beta}^{+} \partial_{\alpha}^{-}) \Psi = 0. \tag{17}$$

Its obvious solution is $\Psi(0, \eta | y^{\pm}) = \Psi(0, \eta | y^{+})$. There is a less obvious solution

$$\widetilde{\Psi}(0 | y^{\pm}) = \epsilon^{\alpha\beta} y_{\alpha}^{-} \partial_{\beta}^{+} \delta(y^{+}). \tag{18}$$

Note that in the case of one variable $x\delta'(x) = -\delta(x)$, but $(y_{\alpha}^{+} \partial_{\beta}^{+} + y_{\beta}^{+} \partial_{\alpha}^{+})\delta(y^{+}) = 0$ in the symplectic case. The latter identity needs to be used to check that $\widetilde{\Psi}$ is a solution. This solution corresponds to the $\Delta = 2$ operator $\bar{\psi}\psi$ that is present in the free fermion theory. It stands out of the main higher-spin current multiplet and we will not discuss it further, except for the two-point function.

Next, we need to recover the $X$-dependence in such a way that $\Psi$ obeys

$$\partial_{\alpha\beta}^{x} \Psi + \tfrac{i}{2} [P_{\alpha\beta}, \Psi]_{\star} = 0. \tag{19}$$

Indeed, the flat space is realised with the help of the gauge function $g = \exp \tfrac{i}{2} X^{\alpha\beta} P_{\alpha\beta}$. The corresponding connection $g^{-1} \star dg = \tfrac{i}{2} dX^{\alpha\beta} P_{\alpha\beta}$, is a flat connection of the conformal algebra. Since $\Psi$ must be in the adjoint representation, the $X$-dependence is determined by

$$\Psi(X, \eta | y^{\pm}) = g^{-1}(X) \star \Psi(0, \eta | y^{+}) \star g(X). \tag{20}$$

This is a direct analog of $O(X) = \exp[X \cdot P]O(0)\exp[-X \cdot P]$ in the standard CFT language, the only difference being that $\Psi$ is a generating function of infinitely many quasi-primary operators. With the help of the oscillator realisation (11), we find

$$g^{-1} \star y_{\gamma}^{+} \star g = y_{\gamma}^{+} + X_{\gamma}{}^{\alpha} y_{\alpha}^{-}, \qquad\qquad g^{-1} \star y_{\gamma}^{-} \star g = y_{\gamma}^{-}. \tag{20}$$

Therefore, the wave-function is constrained now to be

$$\Psi(X, \eta | y^{\pm}) = \Psi(\eta | y_{\gamma}^{+} + X_{\gamma}{}^{\alpha} y_{\alpha}^{-}). \tag{21}$$

Lastly, we need to impose the conservation condition, which determines the $\eta$-dependence

$$\frac{\partial}{\partial X^{\alpha\beta}} \frac{\partial^{2}}{\partial \eta_{\alpha} \partial \eta_{\beta}} \Psi(X, \eta | y^{\pm}) = 0. \tag{22}$$

It is helpful to represent the wave-function as a Fourier integral

$$\Psi(X, \eta | y^{\pm}) = \int d^{2}s \, f(s, \eta) \exp is^{\gamma}[y_{\gamma}^{+} + X_{\gamma}{}^{\alpha} y_{\alpha}^{-}]. \tag{23}$$

Imposing the conservation condition we find

$$\int d^{2}s \, (s_{\alpha} y_{\beta}^{-}) \partial_{\eta}^{\alpha} \partial_{\eta}^{\beta} f(s, \eta) \exp is^{\gamma}[y_{\gamma}^{+} + X_{\gamma}{}^{\alpha} y_{\alpha}^{-}] = 0. \tag{24}$$

Since $f$ cannot depend on $y_{\alpha}^{-}$, otherwise it spoils the solution of the other two conditions, we have to take $f(s, \eta) = f(s^{\gamma} \eta_{\gamma})$. This function of one variable is the expected ambiguity in normalisation of the higher-spin currents. We, of course, fix it to be $f = \exp is^{\gamma} \eta_{\gamma}$. Finally, the wave-function is found to be

$$\Psi(X, \eta | y^{\pm}) = \int d^{2}s \, \exp is^{\gamma}[y_{\gamma}^{+} + X_{\gamma}{}^{\alpha} y_{\alpha}^{-} + \eta_{\gamma}] = \delta^{2}(y_{\gamma}^{+} + X_{\gamma}{}^{\alpha} y_{\alpha}^{-} + \eta_{\gamma}) \equiv \delta^{2}(\Gamma(X)). \tag{25}$$

For completeness, let us note that the Lorentz generators act canonically

$$[L_{\alpha\beta}, \Psi]_\star \big|_{X=0} = \left( y_\alpha^+ \partial_\beta^+ + y_\beta^+ \partial_\alpha^+ \right) \delta^2(y^+ + \eta). \tag{26}$$

Further, the dilation operator relates the conformal weight to the spin

$$[D, \Psi]_\star \big|_{X=0} = \tfrac{1}{2} y^{+\alpha} \partial_\alpha^+ \delta^2(y^+ + \eta). \tag{27}$$

The full higher-spin symmetry can also be seen to act. Its parameters are contained in a covariantly constant generating function of Killing tensors $\xi$, that obeys

$$\partial_{\alpha\beta}^x \xi + \tfrac{i}{2} [P_{\alpha\beta}, \xi]_\star = 0. \tag{28}$$

The conformal and genuine higher-spin symmetries act as $\delta_\xi \Psi = [\Psi, \xi]_\star$.

### 3.3. Correlation Functions

As stated before, the main building block of correlation functions of higher-spin currents is

$$O_n = \mathrm{tr}\left( \Psi_1 \star \cdots \star \Psi_n \right). \tag{29}$$

In order to explicitly compute it, we begin with $O_2$.

#### 3.3.1. Two-Point Correlators

It is useful to first check the two-point functions. Here, we return to the variables $Y^A = (y^\alpha, \bar{y}^\alpha)$ to compute the star-product in what follows. The solution (25) is rewritten as

$$\Psi(X, \eta | Y) = \kappa \, \delta^2(\Gamma(X)), \qquad \text{with } \Gamma^\alpha = A^{\alpha\beta}(X)\bar{y}_\beta + B^{\alpha\beta}(X)y_\beta + C^\alpha, \tag{30}$$

with $C^\alpha = c \, \eta^\alpha$,

$$\begin{aligned}
A^{\alpha\beta}(X) &= a X^{\alpha\beta} - ia \, \epsilon^{\alpha\beta}, \\
B^{\alpha\beta}(X) &= -ia X^{\alpha\beta} + a \, \epsilon^{\alpha\beta} \equiv -iA^{\mathrm{T}}(X) \equiv iA(-X).
\end{aligned} \tag{31}$$

Indeed the most general solution is defined up to a multiplicative constant, $\kappa$, and we can always rescale $y^\pm$ and $\eta$ without affecting Equations (17), (21), and (22). We keep $\kappa$, $a$, and $c$ arbitrary for the time being, in order to derive more general formulas for the star-products and have an additional control over the calculations. Denoting $\Psi_i \equiv \Psi(X_i, \eta_i | Y)$, one can show that

$$\Psi_1 \star \Psi_2 = \frac{-\kappa^2}{|M_{12}|} \exp i \left[ \left( \bar{y} A_2^{\mathrm{T}} + y B_2^{\mathrm{T}} - C_2 \right) \left( M_{12} \right)^{-1} \left( A_1 \bar{y} + B_1 y + C_1 \right) \right], \tag{32}$$

with $M_{12} := A_1 A_2^{\mathrm{T}} + B_1 B_2^{\mathrm{T}}$. With (31), we have $M_{12} = 2ia^2 X_{12}$, so that tracing gives

$$\mathrm{tr}\, \Psi_1 \star \Psi_2 = \frac{-\kappa^2}{4a^4 \, \mathbf{x}_{12}^2} \exp i \left[ \frac{-i}{2a^2} C_{1\alpha} \left( X_{12}^{-1} \right)^{\alpha\beta} C_{2\beta} \right]. \tag{33}$$

In order to match with the two-point function $O_2 = \frac{1}{\mathbf{x}_{12}^2} \exp iP_{12}$ [24], we need

$$\frac{-\kappa^2}{4a^4} = 1, \tag{34}$$

$$\frac{-i}{2a^2} C_{1\alpha} \left( X_{12}^{-1} \right)^{\alpha\beta} C_{2\beta} = \eta_{1\alpha} (X_{12}^{-1})^{\alpha\beta} \eta_{2\beta}. \tag{35}$$

With the choice $C_i = c\,\eta_i$, the second equation leads to $\dfrac{i\,c^2}{2a^2} = -1$. In particular, our wave-function $\Psi$ leads to the correct two-point function. Let us note that

$$\Psi_{12} := \Psi_1 \star \Psi_2 = \frac{1}{\mathbf{x}_{12}^2}\,\exp i\left[\tfrac{1}{2}\Sigma_{12}^{AB}\,Y_A Y_B + \xi_{12}^A\,Y_A + P_{12}\right] \tag{36}$$

with

$$\Sigma_{12}^{AB} = \frac{i}{a^2}\begin{pmatrix} B_2^{\mathsf{T}} X_{12}^{-1} B_1 & B_2^{\mathsf{T}} X_{12}^{-1} A_1 \\ A_2^{\mathsf{T}} X_{12}^{-1} B_1 & B_2^{\mathsf{T}} X_{12}^{-1} A_1 \end{pmatrix}^{(AB)}, \tag{37}$$

$$\xi_{12}^A = \frac{i}{2a^2}\begin{pmatrix} c\,\eta_1 & c\,\eta_2 \end{pmatrix}^B \begin{pmatrix} X_{12}^{-1} B_2 & X_{12}^{-1} A_2 \\ X_{12}^{-1} B_1 & X_{12}^{-1} A_1 \end{pmatrix}^A_{\;B} \equiv \eta_{12}^B\,\rho_{12B}{}^A. \tag{38}$$

We will denote the Gaussian (36) as $\Phi(\Sigma_{12}, \xi_{12}, \theta_{12})$. Crucial properties to compute higher-order correlators are $(\Sigma_{12}^2)^{AB} \equiv (\Sigma_{12})^{AC}(\Sigma_{12})_C{}^B = \epsilon^{AB}$ and $\Sigma_{12} = -\Sigma_{21}$.

Let us also check that the second solution, (18), leads to the correct two-point function. The solution can be rewritten as

$$\widetilde{\Psi}(X|y^{\pm}) = \frac{\partial}{\partial\chi}\delta^2\big(y_\alpha^+ + X_\alpha{}^\beta y_\beta^- + \chi y_\alpha^-\big)\Big|_{\chi=0}. \tag{39}$$

In terms of $(y_\alpha, \bar{y}_\alpha)$, we write the argument of the delta function as $\widetilde{\Gamma}(X) = \widetilde{A}(X)\bar{y} + \widetilde{B}(X)y$, with

$$\widetilde{A}^{\alpha\beta}(X) = bX^{\alpha\beta} + b(\chi - i)\,\epsilon^{\alpha\beta}, \qquad \widetilde{B}^{\alpha\beta}(X) = -ibX^{\alpha\beta} + b(1 - i\chi)\,\epsilon^{\alpha\beta}, \tag{40}$$

where we introduced an arbitrary constant $b$. With those notations, we can use the previous result (32) with $C_i \to 0$, $\kappa \to \widetilde{\kappa}$ and $A, B \to \widetilde{A}, \widetilde{B}$ to get

$$\widetilde{\Psi}(X_1|Y) \star \widetilde{\Psi}(X_2|Y) = \widetilde{\kappa}^2\,\frac{\partial^2}{\partial\chi_1\partial\chi_2}\left[\frac{1}{|\widetilde{M}_{12}|}\,\exp i\left[\tfrac{1}{2}\,\widetilde{\Sigma}_{12}^{AB}\,Y_A Y_B\right]\right]\Bigg|_{\chi_1 = \chi_2 = 0}, \tag{41}$$

with $\widetilde{M}_{12} = 2ib^2\,X_{12} + (\chi_1 + \chi_2)\epsilon$ and

$$\widetilde{\Sigma}_{12}^{AB} = 2\begin{pmatrix} \widetilde{B}_2^{\mathsf{T}} \widetilde{M}_{12}^{-1} \widetilde{B}_1 & \widetilde{B}_2^{\mathsf{T}} \widetilde{M}_{12}^{-1} \widetilde{A}_1 \\ \widetilde{A}_2^{\mathsf{T}} \widetilde{M}_{12}^{-1} \widetilde{B}_1 & \widetilde{B}_2^{\mathsf{T}} \widetilde{M}_{12}^{-1} \widetilde{A}_1 \end{pmatrix}^{(AB)}. \tag{42}$$

Tracing gives

$$\operatorname{tr}\widetilde{\Psi}_1 \star \widetilde{\Psi}_2 = \frac{\widetilde{\kappa}^2}{2\,b^4}\frac{1}{\mathbf{x}_{12}^4}, \tag{43}$$

which is the two-point function of the $\Delta = 2$ operator, which in our case is $\bar{\psi}\psi$.

### 3.3.2. Higher-Point Procedure

Having the building block, we can now describe the procedure to obtain $O_n$. We begin with $O_{2n}$ $(n \in \mathbb{N}_0)$. Since the star-product is associative, we can compute it recursively as

$$O_{2n} = \operatorname{tr}\Big((\Psi_{1,2} \star \Psi_{3,4} \star \cdots \star \Psi_{2n-3,2n-2}) \star \Psi_{2n-1,2n}\Big). \tag{44}$$

Since we know that $\Psi_{i,j}$ is a Gaussian in $Y^A$, we would have to compute the star-product of Gaussians. This will be performed below, but we already note that the star-product of Gaussians is a Gaussian. For the $(2n+1)$-point correlator, we have

$$O_{2n+1} = \operatorname{tr}\left(\left(\Psi_{1,2} \star \Psi_{3,4} \star \cdots \star \Psi_{2n-1,2n}\right) \star \Psi_{2n+1}\right). \tag{45}$$

Since Gaussians form a closed subalgebra under the star-product, the term in the small parentheses is a Gaussian of the form $r(\mathbf{x})\,\Phi(\Sigma, \xi, \theta)$, and one can show that

$$\Phi(\Sigma, \xi, \theta) \star \Psi_k = \frac{i\kappa\,\Phi(\Sigma, \xi, \theta)}{\sqrt{|-\gamma_k \Sigma \gamma_k^{\mathrm{T}}|}} \exp \frac{i}{2} b \left(\gamma_k \Sigma \gamma_k^{\mathrm{T}}\right)^{-1} b \tag{46}$$

with

$$\gamma_k^{\alpha B} = \left(B_k^{\alpha\beta} \quad A_k^{\alpha\dot\beta}\right), \qquad b^A = \left(\gamma_k Y - \gamma_k \Sigma Y - \gamma_k \xi + C_k\right)^A. \tag{47}$$

We also note that $\left(\gamma_k \Sigma \gamma_k^{\mathrm{T}}\right)^{-1} = -\left(\gamma_k \Sigma \gamma_k^{\mathrm{T}}\right)/|\gamma_k \Sigma \gamma_k^{\mathrm{T}}|$. In the following, we denote

$$\Psi_{1,2} \star \cdots \star \Psi_{n-1,n} \sim \exp i\left[\frac{1}{2}\left(\Sigma_{[n]}\right)_{AB} Y^A Y^B + \xi_{[n]}^A Y_A + \theta_{[n]}\right]. \tag{48}$$

Therefore, taking $Y = 0$, the argument of the exponential for (45) reads

$$\theta_{[2n+1]} = \theta_{[2n]} - \frac{1}{2}\left(\xi_{[2n]}\gamma_{2n+1}^{\mathrm{T}} + \eta_{2n+1}\right) \frac{\gamma_{2n+1}\,\Sigma_{1,2n}\,\gamma_{2n+1}^{\mathrm{T}}}{|\gamma_{2n+1}\,\Sigma_{1,2n}\,\gamma_{2n+1}^{\mathrm{T}}|}\left(-\gamma_{2n+1}\xi_{[2n]} + \eta_{2n+1}\right). \tag{49}$$

This suggests that we first need to compute the $2n$-point correlators. Let us see how the star-product of Gaussians works.

### 3.3.3. Star-Product of Gaussians

Let us be more general and consider a Gaussian

$$\Phi(f, \xi, \theta) = \exp i\left[\frac{1}{2} f_{AB} Y^A Y^B + \xi^A Y_A + \theta\right], \quad A, B = 1, \ldots, 2N, \tag{50}$$

with $f_{AB}$ a symmetric matrix, $\xi^A$ a commuting spinor, and $\theta$ a constant. One can show that [24,63]

$$\Phi(f, \xi, 0) \star \Phi(g, \eta, 0) = \frac{(-1)^N}{\sqrt{|f|\,|g + f^{-1}|}}\,\Phi(f \circ g, \xi \circ \eta, q) \tag{51}$$

where (matrix 1 in (52) should be understood as $\epsilon_{AB}$)

$$\begin{aligned}
(f \circ g)_{AB} &= \frac{1}{1 + gf}(1 + g) - \frac{1}{1 + fg}(1 - f), \\
(\xi \circ \eta)^A &= \xi^B\left[\frac{1}{1 + gf}(1 + g)\right]_B^{\;A} + \eta^B\left[\frac{1}{1 + fg}(1 - f)\right]_B^{\;A}, \\
q &= \frac{1}{2}\left(\frac{1}{1 + gf}\,g\right)_{AB}\xi^A \xi^B + \frac{1}{2}\left(\frac{1}{1 + fg}\,f\right)_{AB}\eta^A \eta^B \\
&\quad - \left(\frac{1}{1 + gf}\right)_{AB}\xi^A \eta^B.
\end{aligned} \tag{52}$$

The proof follows from Gaussian integration. In our $\Psi_{i,j}$, (36), we have $f^2 = \epsilon = g^2$. In that case, one has

$$
\begin{aligned}
(f \circ g)_{AB} &= \frac{1}{f+g}(2+g-f), \\
(\xi \circ \eta)^A &= \frac{1}{2}\xi^B (1+f \circ g)_B{}^A + \frac{1}{2}\eta^B (1 - f \circ g)_B{}^A, \\
q &= \frac{1}{8}\{f,g\}_{\circ AB}\left(\xi^A \xi^B + \eta^A \eta^B\right) - \frac{1}{2}\left(1 + \frac{1}{2}[f,g]_\circ\right)_{AB}\xi^A \eta^B,
\end{aligned}
\tag{53}
$$

and $|f||g+f^{-1}| = |f+g|$. We note that $(f \circ g)_{AB}$ is symmetric.

### 3.3.4. Higher-Point Correlators

Now that we have simple formulas, we can proceed to compute (44), whose building block is $\Psi_{i,j} \star \Psi_{k,l}$. A crucial property of our $\Sigma$ matrices is that

$$
\Sigma_{ij} \circ \Sigma_{kl} = \Sigma_{il}.
\tag{54}
$$

This tells us that the only matrices $f$, $g$ and $f \circ g$ that appear in (53) are our $\Sigma$. Therefore, it is useful to use the projectors $\pi_{ij}^{\pm} := \frac{1}{2}(\epsilon \pm \Sigma_{ij})$ that satisfy the following properties:

$$
\begin{aligned}
\left(\pi_{ij}^{\pm}\right)^{\mathrm{T}} &= -\pi_{ij}^{\mp}, & \pi_{ji}^{\pm} &= \pi_{ij}^{\mp}, \\
\pi_{ij}^{\pm}\pi_{ij}^{\mp} &= 0, & \pi_{ij}^{\pm}\pi_{ij}^{\pm} &= \pi_{ij}^{\pm}, \\
\pi_{ij}^{-}\pi_{ik}^{+} &= 0, & \pi_{ij}^{+}\pi_{ik}^{+} &= \pi_{ik}^{+}.
\end{aligned}
\tag{55}
$$

Now we proceed recursively. For the two-point correlator, we had (36). For the four-point one, writing (53) in terms of projectors, we have

$$
\begin{aligned}
\Sigma_{[4]} &:= \Sigma_{12} \circ \Sigma_{34} \equiv \Sigma_{14}, \\
\xi_{[4]} &:= \xi_{12} \circ \xi_{34} = \xi_{12}^B\left(\pi_{14}^+\right)_B{}^A + \xi_{34}^B\left(\pi_{14}^-\right)_B{}^A, \\
\theta_{[4]} &= P_{12} + P_{34} + \frac{1}{4}\left(\pi_{14}^+ - \pi_{23}^+\right)_{AB}\left(\xi_{12}^A\xi_{12}^B + \xi_{34}^A\xi_{34}^B\right) - \frac{1}{2}\left(\pi_{14}^+ + \pi_{23}^+\right)_{AB}\xi_{12}^A\xi_{34}^B.
\end{aligned}
\tag{56}
$$

Thanks to the structures listed in Appendix B, one obtains

$$
\theta_{[4]} = \frac{1}{2}\left[P_{12} + P_{23} + P_{34} - P_{41}\right] + \frac{1}{4}\left[Q_{24}^1 + Q_{31}^2 + Q_{42}^3 + Q_{13}^4\right].
\tag{57}
$$

The properties of the projectors (55) help us to easily generalise

$$
\begin{aligned}
\Sigma_{[2n]} &= \Sigma_{1,2n}, \\
\xi_{[2n]}^A &= \xi_{12}^B\left(\pi_{1,2n}^+\right)_B{}^A + \xi_{2n-1,2n}^B\left(\pi_{1,2n}^-\right)_B{}^A, \\
\theta_{[2n]} &= \theta_{[2n-2]} + \frac{1}{4}\left(\pi_{1,2n}^+ - \pi_{2n-2,2n-1}^+\right)_{AB}\left(\xi_{[2n-2]}^A \xi_{[2n-2]}^B + \xi_{2n-1,2n}^A \xi_{2n-1,2n}^B\right) \\
&\quad - \frac{1}{2}\left(\pi_{1,2n}^+ + \pi_{2n-2,2n-1}^+\right)_{AB}\xi_{[2n-2]}^A\xi_{2n-1,2n}^B + P_{2n-1,2n}.
\end{aligned}
\tag{58}
$$

One finds

$$
\begin{aligned}
\theta_{[2n]} &= \theta_{[2n-2]} + \frac{1}{4}\left[Q_{2n-2,2n}^1 + Q_{2n-1,1}^{2n-2} + Q_{2n,2n-2}^{2n-1} + Q_{1,2n-1}^{2n}\right] \\
&\quad + \frac{1}{2}\left[P_{1,2n} - P_{1,2n-2} + P_{2n-2,2n-1} + P_{2n-1,2n}\right],
\end{aligned}
\tag{59}
$$

which reduces to

$$\theta_{[2n]} = \frac{1}{4} \sum_{i=1}^{2n} Q^i_{i+1,i-1} + \frac{1}{2} \sum_{i=1}^{2n} (-)^{\delta_{i,2n}} P_{i,i+1} \,, \tag{60}$$

where the sum is understood to be mod $2n$. For prefactors, we have $|f + g|^{-1/2}$ from (51), but one should also take into account the prefactor of every $\Psi_{i,j}$, e.g., $x_{12}^{-2}$. Due to

$$|\Sigma_{ij} + \Sigma_{kl}| = 16 \frac{|X_{jk}||X_{il}|}{|X_{ij}||X_{kl}|} \,, \tag{61}$$

and requiring (34), one can find the prefactor for $O_{2n}$

$$\left[ 2^{2n-2} \prod_{i=1}^{2n} x_{i,i+1} \right]^{-1}. \tag{62}$$

3.3.5. $(2n+1)$-pt Functions

We can now compute (45). With the help of (49) and thanks to the structures in Appendix B, one has

$$\begin{aligned}
\theta_{[2n+1]} &= \theta_{[2n]} + \frac{1}{4} \left( Q^1_{2n,2n+1} + Q^{2n}_{2n+1,1} + Q^{2n+1}_{1,2n} \right) + \frac{1}{2} \left( P_{1,2n+1} + P_{2n,2n+1} - P_{1,2n} \right) \\
&= \frac{1}{4} \sum_{i=1}^{2n+1} Q^i_{i+1,i-1} + \frac{1}{2} \sum_{i=1}^{2n+1} (-)^{\delta_{i,2n+1}} P_{i,i+1} \,.
\end{aligned} \tag{63}$$

From (46) and with $|\gamma_k \Sigma_{ij} \gamma_k^{\mathrm{T}}| = -16a^4 |X_{ik}||X_{jk}|/|X_{ij}|$, the prefactor is found to be (the sign of $\kappa$ can be taken as $-1$ to obtain a positive function)

$$\left[ 2^{2n-1} \prod_{i=1}^{2n+1} x_{i,i+1} \right]^{-1}. \tag{64}$$

Therefore, we can summarise the results for any $n \in \mathbb{N}_0$ by

$$O_n = \frac{1}{2^{n-2} \prod_{i=1}^{n} x_{i,i+1}} \exp i \left[ \frac{1}{4} \sum_{i=1}^{n} Q^i_{i+1,i-1} + \frac{1}{2} \sum_{i=1}^{n} (-)^{\delta_{i,n}} P_{i,i+1} \right], \tag{65}$$

where the sum and the product are understood to be mod $n$.

## 4. Conclusions and Discussion

The main results of the paper are the wave-functions that represent higher-spin currents multiplets in the higher-spin algebra, and the calculation of correlators (65). The result (65) is exactly the conformally invariant generating function of correlators found in [23,24,26]. Insertions of the $\Delta = 2$ wave-function $\widetilde{\Psi}$, (18), will have to lead to the results of [25]. It should be noted that the wave-functions on the CFT side found in this paper are much simpler than those on $AdS_4$ of [23–26]. We hope that the CFT wave-functions provide a useful first step to compute the deformed invariants [33] of the slightly-broken higher-spin symmetry.

Lastly, it is also worth mentioning recent works [64–66] that apply similar ideas to directly looking for higher-spin invariant observables. The higher-spin algebra of these papers is a commutative limit of the higher-spin algebra we use in this paper along two (out of four) oscillators. In particular, the amplitudes (and wave-functions) of [66] should be some 'flat limits' of the correlation functions of the present paper. It would also be interesting to extend the results of this paper to other dimensions and to find the CFT counterpart of the rather complicated bulk-to-boundary propagator found in [67].

**Funding:** This research was supported by the European Research Council (ERC) under the European Union's Horizon 2020 research and innovation programme (grant agreement no. 101002551).

**Institutional Review Board Statement:** Not applicable.

**Informed Consent Statement:** Not applicable.

**Acknowledgments:** This work is a part of a Master's thesis, defended at University of Mons (Belgium) in June 2022 under the supervision of E. Skvortsov. The author is grateful to E. Skvortsov for many stimulating discussions.

**Conflicts of Interest:** The author declares no conflict of interest.

## Appendix A. Vector-Spinor Dictionary

In $3d$, the Lorentz algebra is $so(2,1)$, isomorphic to $sl(2,\mathbb{R})$. From this fact, any Lorentz vector can be represented as a $2 \times 2$ symmetric matrix. Indeed, if $\{\mathbf{x}^m\}_{m=0,1,2}$ are the components of a 3-vector $\mathbf{x}$, with the following Pauli matrices

$$(\sigma_0)^{\alpha\beta} = \begin{pmatrix} 1 & 0 \\ 0 & 1 \end{pmatrix}, \qquad (\sigma_1)^{\alpha\beta} = \begin{pmatrix} 1 & 0 \\ 0 & -1 \end{pmatrix}, \qquad (\sigma_2)^{\alpha\beta} = \begin{pmatrix} 0 & 1 \\ 1 & 0 \end{pmatrix}, \qquad (A1)$$

we can form the matrix $X$ of components $X^{\alpha\beta} = \mathbf{x}^m (\sigma_m)^{\alpha\beta}$. A Lorentz transformation corresponds to an SL$(2,\mathbb{R})$ matrix $A_\alpha{}^\beta$ acting as $X^{\alpha\beta} \to X^{\gamma\delta} A_\gamma{}^\alpha A_\delta{}^\beta$. For $\mathbf{x}^m = (t, x, y)$, we have

$$X^{\alpha\beta} = \begin{pmatrix} t+x & y \\ y & t-x \end{pmatrix} \qquad \text{and} \qquad (X^{-1})^{\alpha\beta} = \frac{-1}{|X|} X^{\alpha\beta}, \qquad (A2)$$

where the determinant $|X|$ is $-|\mathbf{x}|^2 = -\eta_{mn}\mathbf{x}^m\mathbf{x}^n$, since we take $\eta_{mn}$ as the Minkowski metric of signature $(-++)$. We observe that $X^{-1}$ is obtained from an inversion $R\mathbf{x}^m = \mathbf{x}^m/|\mathbf{x}|^2$, i.e., $RX^{\alpha\beta} = (X^{-1})^{\alpha\beta}$. We also note that

$$\frac{\partial}{\partial X^{\alpha\beta}} X^{\gamma\delta} := \partial_{\alpha\beta} X^{\gamma\delta} = \frac{1}{2}\left(\delta_\alpha{}^\gamma \delta_\beta{}^\delta + \delta_\alpha{}^\delta \delta_\beta{}^\gamma\right). \qquad (A3)$$

We introduce the SL$(2,\mathbb{R})$-invariant tensor $\epsilon_{\alpha\beta} = -\epsilon_{\beta\alpha}$, with $\epsilon_{12} = +1$ and its inverse, such that $\epsilon_{\alpha\beta}\epsilon^{\gamma\beta} = \delta_\alpha{}^\gamma$, i.e., $\epsilon^{12} = +1$. With them, one can raise and lower spinorial indices. For a spinor $\xi^\alpha$, we use Penrose's conventions:

$$\xi_\alpha = \xi^\beta \epsilon_{\beta\alpha}, \qquad \xi^\alpha = \epsilon^{\alpha\beta} \xi_\beta. \qquad (A4)$$

We also define $\partial_\alpha \equiv \frac{\partial}{\partial \xi^\alpha}$, such that $\partial_\alpha \xi_\beta = \epsilon_{\alpha\beta}$ and $\partial^\alpha = \epsilon^{\alpha\beta}\partial_\beta$. The contraction between two spinors $\chi, \xi$ is defined as $\chi\xi \equiv \chi^\alpha\xi_\alpha = -\xi\chi$. Finally, we note that any bi-spinor $A_{\alpha\beta}$ can be written as

$$A_{\alpha\beta} = S_{\alpha\beta} + \tfrac{1}{2} A_\lambda{}^\lambda \epsilon_{\alpha\beta} \qquad (A5)$$

with $S_{\alpha\beta} = S_{\beta\alpha}$ and $A_\lambda{}^\lambda = \epsilon^{\lambda\gamma} A_{\lambda\gamma}$ the symplectic trace of $A_{\alpha\beta}$. In addition, we write the matrix multiplication between two matrices $A$ and $B$ as $(AB)_\alpha{}^\gamma \equiv A_\alpha{}^\beta B_\beta{}^\gamma$ and $(A\xi)^\alpha \equiv A^{\alpha\beta}\xi_\beta$. Then, $\mathbf{x}^m = \frac{1}{2}(X\sigma^m)_\alpha{}^\alpha$.

## Appendix B. Conformal Structures

We first recall the notation we already introduced, (38),

$$\eta_{ij} = c\begin{pmatrix} \eta_i^A & \eta_j \end{pmatrix}^A, \qquad\qquad \rho_{ij}^{AB} = \frac{i}{2a^2}\begin{pmatrix} X_{12}^{-1}B_2 & X_{12}^{-1}A_2 \\ X_{12}^{-1}B_1 & X_{12}^{-1}A_1 \end{pmatrix}^{AB}. \qquad (A6)$$

For $2n$-pt functions, we needed the following structures (we always have a factor $\frac{-ic^2}{2a^2}$ but, due to (35), this factor is 1)

$$\eta^A_{ij}\, \eta^B_{ij}\, \left[\rho_{ij}\, \pi^+_{ik}\, \rho^{\mathrm{T}}_{ij}\right]_{AB} = Q^i_{kj} + P_{ij}, \qquad \eta^A_{ij}\, \eta^B_{ij}\, \left[\rho_{ij}\, \pi^+_{jk}\, \rho^{\mathrm{T}}_{ij}\right]_{AB} = Q^j_{ki} - P_{ij}, \tag{A7}$$

$$\eta^A_{ij}\, \eta^B_{ij}\, \left[\rho_{ij}\, \pi^+_{ik}\, \pi^-_{il}\, \rho^{\mathrm{T}}_{ij}\right]_{AB} = Q^i_{kl}, \qquad \eta^A_{ij}\, \eta^B_{ij}\, \left[\rho_{ij}\, \pi^+_{jk}\, \pi^-_{jl}\, \rho^{\mathrm{T}}_{ij}\right]_{AB} = Q^j_{kl}, \tag{A8}$$

$$\eta^A_{ij}\, \eta^B_{kl}\, \left[\rho_{ij}\, \pi^+_{il}\, \rho^{\mathrm{T}}_{kl}\right]_{AB} = P_{il}, \qquad \eta^A_{ij}\, \eta^B_{kl}\, \left[\rho_{ij}\, \pi^+_{jk}\, \rho^{\mathrm{T}}_{kl}\right]_{AB} = P_{jk}. \tag{A9}$$

For $(2n+1)$-pt functions, with $f := (\gamma^{\mathrm{T}}\gamma)_{2n+1}\Sigma_{1,2n}(\gamma^{\mathrm{T}}\gamma)_{2n+1}/|\gamma\Sigma_{1,2n}\gamma|$,

$$\eta^A_{12}\, \eta^B_{12}\, \left[\rho_{12}\, \pi^+_{1,2n}\, f\, \pi^-_{1,2n}\, \rho^{\mathrm{T}}_{12}\right]_{AB} = \tfrac{1}{2} Q^1_{2n,2n+1}, \tag{A10}$$

$$\eta^A_{2n-1,2n}\, \eta^B_{2n-1,2n}\, \left[\rho_{2n-1,2n}\, \pi^-_{1,2n}\, f\, \pi^+_{1,2n}\, \rho^{\mathrm{T}}_{2n-1,2n}\right]_{AB} = -\tfrac{1}{2} Q^{2n}_{1,2n+1}, \tag{A11}$$

$$\eta^A_{12}\, \eta^B_{2n-1,2n}\, \left[\rho_{12}\, \pi^+_{1,2n}\, f\, \pi^+_{1,2n}\, \rho^{\mathrm{T}}_{2n-1,2n}\right]_{AB} = -\tfrac{1}{2} P_{1,2n}, \tag{A12}$$

and, with $g := (\gamma^{\mathrm{T}}\gamma)_{2n+1}\Sigma_{1,2n}\gamma^{\mathrm{T}}_{2n+1}$,

$$\eta^A_{12}\, \eta^\beta_{2n+1}\, \left[\rho_{12}\, \pi^+_{1,2n}\, g\right]_{A\beta} = -\tfrac{1}{2} P_{1,2n+1}, \tag{A13}$$

$$\eta^A_{2n-1,2n}\, \eta^\beta_{2n+1}\, \left[\rho_{2n-1,2n}\, \pi^-_{1,2n}\, g\right]_{A\beta} = -\tfrac{1}{2} P_{2n,2n+1}. \tag{A14}$$

Lastly,

$$\eta^\alpha_{2n+1}\, \eta^\beta_{2n+1}\, \left[\frac{\gamma_{2n+1}\Sigma_{1,2n}\gamma^{\mathrm{T}}_{2n+1}}{|\gamma\Sigma\gamma|}\right]_{\alpha\beta} = \tfrac{1}{2} Q^{2n+1}_{1,2n}. \tag{A15}$$

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
