# Peer review of "On Correlation Functions as Higher-Spin Invariants"

_symmetry, doi:10.3390/sym15040950_

Round 1

Reviewer 1 Report

In the present manuscript the Author streamlines the analysis of [24] by deriving direct wave functions that connect the usual spinor variables of the AdS_4 higher-spin algebra to the CFT adapted variables. This allows the Author to reproduce the known higher-spin invariant CFT correlation functions in a simpler way. These results can be used to streamline other related computations. I recommend this manuscript for publication. Though, I also recommend the Author to consider the following corrections.

1) From the abstract and the introduction it is not quite clear what the Author is going to do in the following. In particular the Author devotes quite a lot of discussions to slightly broken higher-spin algebras and strongly homotopy algebras, which are, in fact, irrelevant for the rest of the text. The second sentence of the abstract is precisely like that. So, I invite the Author to remove the unnecessary discussion on the slightly broken case from the abstract and, probably, reduce them in the introduction. Instead, it should be explained more clearly what the Author actually does. In particular, "exclude the holographic aspect" on line 53 is not precise enough to be able to understand what it means.

2) K in (16) were not explicitly given. It is also not quite clear whether it is the same K that appears in (30) and in what follows.

3) Normally, one would expect an ambiguity in the definition of $\Psi$ related to the fact that $\delta(y)\star\delta(\bar y)\star \Psi$ transform the same way as $\Psi$ does. Could the Author clarify this point? 

4) The logic of a passage from (25) to (30) is not quite clear. It seems that $\Psi$ was already derived in (25). Why can one deform it with an extra factor $K$ in front, extra parameters $a$, $c$ and $\epsilon$. Besides that $\epsilon$ was not defined and does not seem to feature any formulas after (31). This should be better explained. 

5) In line 193, there is $(\Sigma_{12}^2)^{AB}$. It is not quite clear what the upper index 2 refers to. If it means squared, it would be better to write an explicit formula to see how the $A,B$ indices are contracted.

Author Response

Thanks for your comments and suggestions.

1) From the abstract and the introduction it is not quite clear what the Author is going to do in the following. In particular the Author devotes quite a lot of discussions to slightly broken higher-spin algebras and strongly homotopy algebras, which are, in fact, irrelevant for the rest of the text. The second sentence of the abstract is precisely like that. So, I invite the Author to remove the unnecessary discussion on the slightly broken case from the abstract and, probably, reduce them in the introduction. Instead, it should be explained more clearly what the Author actually does. In particular, "exclude the holographic aspect" on line 53 is not precise enough to be able to understand what it means.

We agree that the abstract should be modified to stress more the actual results. Accordingly, we removed the sentence about strong homotopy algebras. We have also reduced the discussion of the slightly-broken case in the introduction, but we feel that a brief overview of this story is important to understand the more general problem and the motivation behind the paper. We have also rearranged the text in the last but one paragraph to explain better what "exclude the holographic aspect" means.

2) K in (16) were not explicitly given. It is also not quite clear whether it is the same K that appears in (30) and in what follows.

Indeed, the exact form of K and F are not given in (16) as their form is not needed in the following text. This is only meant to outline class of functions used in the holographic calculations. We clarify this point more now around (16). We also changed K in (30) and below to kappa to avoid any clash in notation. 

3) Normally, one would expect an ambiguity in the definition of $\Psi$ related to the fact that $\delta(y)\star\delta(\bar y)\star \Psi$ transform the same way as $\Psi$ does. Could the Author clarify this point? 

This is an interesting point. Since the transformation is discrete and squares to one, it can be related to the inversion symmetry of the setup. At least the naive application of the Fourier transform swaps $y^+$ and $y^-$, which also exchanges $P_{\alpha\beta}$ and $K^{\alpha\beta}$. At the moment, we are not sure how to use this observation. 

4) The logic of a passage from (25) to (30) is not quite clear. It seems that $\Psi$ was already derived in (25). Why can one deform it with an extra factor $K$ in front, extra parameters $a$, $c$ and $\epsilon$. Besides that $\epsilon$ was not defined and does not seem to feature any formulas after (31). This should be better explained. 

In (30) and below we define a slightly more general wave-function to derive more general formulas for the star-products that are used below and to also have an additional control over the result, for example, to see that only for the right wave-functions the correlators are correct. We hope to explain this better around (30). We have also changed (31) to show the index structure, where $\epsilon\equiv \epsilon^{\alpha\beta}$.

5) In line 193, there is $(\Sigma_{12}^2)^{AB}$. It is not quite clear what the upper index 2 refers to. If it means squared, it would be better to write an explicit formula to see how the $A,B$ indices are contracted.

We explained the notation there. Indeed, it is the square of a matrix.

Reviewer 2 Report

This is a detailed paper which is deriving correlation functions as higher spin invariants of CS vector models. The new derivation would be of interest to researchers on this topic as it reproduces earlier results, and also shows a new approach. This manuscript may be published, subject to other requirements, in the journal SYMMETRY (MDPI).  

Author Response

Thank you for your comments.

Reviewer 3 Report

I have gone through the whole manuscript again titled on “On correlation functions as higher-spin invariants”.

Overall, the presentation of the article looks very good and English grammar is also good.   The manuscript is potentially publishable, but it requires some minor revisions, please see the point below to be addressed. I recommend the article for publication after minor revisions.

Minor revisions are required

1.      The citations are not in sequential order. 

2.      The conclusion section is missing.

Author Response

Thank you for your comments.

  1. The citations are not in sequential order. 

Perhaps, the wrong order of citation was a software bug because it now complies correctly. The references are numbered in the order of their appearance.

  1. The conclusion section is missing.

We have a short discussion section, which is not slightly expanded and turned into 'Conclusions and Discussion'.